# Prediction of Phage Virion Proteins Using Machine Learning Methods

**DOI:** 10.3390/molecules28052238

**Published:** 2023-02-28

**Authors:** Ranjan Kumar Barman, Alok Kumar Chakrabarti, Shanta Dutta

**Affiliations:** 1Division of Virology, ICMR-National Institute of Cholera and Enteric Diseases, P-33, C.I.T.Road Scheme XM, Beliaghata, Kolkata 700010, West Bengal, India; 2Division of Bacteriology, ICMR-National Institute of Cholera and Enteric Diseases, P-33, C.I.T.Road Scheme XM, Beliaghata, Kolkata 700010, West Bengal, India

**Keywords:** AMR, bacteriophage, phage virion protein, machine learning, phage therapy, web server

## Abstract

Antimicrobial resistance (AMR) is a major problem and an immediate alternative to antibiotics is the need of the hour. Research on the possible alternative products to tackle bacterial infections is ongoing worldwide. One of the most promising alternatives to antibiotics is the use of bacteriophages (phage) or phage-driven antibacterial drugs to cure bacterial infections caused by AMR bacteria. Phage-driven proteins, including holins, endolysins, and exopolysaccharides, have shown great potential in the development of antibacterial drugs. Likewise, phage virion proteins (PVPs) might also play an important role in the development of antibacterial drugs. Here, we have developed a machine learning-based prediction method to predict PVPs using phage protein sequences. We have employed well-known basic and ensemble machine learning methods with protein sequence composition features for the prediction of PVPs. We found that the gradient boosting classifier (GBC) method achieved the best accuracy of 80% on the training dataset and an accuracy of 83% on the independent dataset. The performance on the independent dataset is better than other existing methods. A user-friendly web server developed by us is freely available to all users for the prediction of PVPs from phage protein sequences. The web server might facilitate the large-scale prediction of PVPs and hypothesis-driven experimental study design.

## 1. Introduction

Bacteriophages or phages are viruses that can infect and kill pathogenic bacteria. Phages were discovered a century ago by Frederick William Twort in 1915 and Félix d’Hérelle in 1917 independently. They are among the most abundant entities in the universe and can be found in diverse environments wherever bacteria are abundant, including water, soil, and inside plants and animals [1,2]. The impact of bacteriophages in biomedical research as a model organism to facilitate the birth of molecular biology and its contribution to strengthening molecular biology research has been reviewed extensively [3]. Since its discovery, over the last 100 years, several attempts have been made to use phages to treat bacterial infections. However, due to a lack of proper knowledge and other logistical problems, they have never been successfully used to control bacterial infections in the past. The discovery of antibiotics in the 1928s and the widespread acceptance of antibiotics gradually suppressed the use and interest in phage therapy. Antibiotics have been used to treat bacterial diseases to control a wide range of bacterial infections; however, the improper use of antibiotics led to the gradual emergence of drug resistance in bacteria. Recent emergence of antimicrobial resistance challenges existing treatment regimens since the pipeline of new drug discovery has apparently dried up over the last couple of decades. Bacteriophages, as biocontrol agents, are considered as one of the best choices to fight against multidrug-resistant bacteria. Hence, the use of bacteriophages has regained a special interest almost 100 years after their discovery [4,5]. Special characteristics, including nontoxicity, harmlessness to humans, and natural killers of pathogenic bacteria, have made phages and their proteins as an alternatives to antibiotics [6]. Therefore, the scientific community is searching for novel phages and their proteins as therapeutic targets [7]. Mainly, two groups of proteins are found in phages, virion (structural) and non-virion (nonstructural). Phage virion proteins (PVPs) are attached to the surface of the host and implant genetic materials within the host cell. The identification of PVPs not only helps to understand the relationship between the phage and host but also facilitates the development of novel antibacterial drugs [8]. Several experimental techniques including protein array, gel electrophoresis, and mass spectrometry have been used for the identification of PVPs and non-PVPs [9,10,11]. However, these standard experimental techniques are very expensive and extremely laborious and time consuming, which led us to find an easier way to identify PVPs and non-PVPs. Recent advancements in the development of computational methods have contributed significantly to the identification of PVPs and non-PVPs. These methods have led to the development of a computational method for the identification of PVPs and non-PVPs.

Several computational methods have been proposed to predict PVPs and non-PVPs using protein sequence features [12,13,14,15,16]. Feng et al. introduced a Naïve Bayes-based method to predict PVPs and non-PVPs using protein primary sequence features such as amino acid composition (AAC) and dipeptide composition (DPC) [14]. The authors achieved an accuracy of 79.15% with a sensitivity of 75.76% and a specificity of 80.77% on a training dataset. However, the performance of their method on an independent dataset was not mentioned and they have not developed a web server to predict PVPs for users. The same group also proposed another computational method to predict PVPs using g-gap DC features [13]. They used the analysis of variance (ANOVA) with incremental feature selection (IFS) techniques and achieved an accuracy of 85.02% and 71.3% on training and independent datasets, respectively. They have developed a web server to predict PVPs and non-PVPs for any user. However, currently, the web server is not working. Tan et al. proposed a similar type of method, which employed ANOVA and the minimal-redundancy-maximal-relevance (mRMR) with IFS for the identification of PVPs [16]. This method attained an accuracy of 87.95% and 75.53% on training and independent datasets, respectively. However, a web server to predict PVPs for any user has not been developed. Manavalan et al. introduced a support vector machine (SVM)-based method to predict PVPs and non-PVPs using protein sequence composition features [15]. This method achieved an accuracy of 87% and 79.8% on training and independent datasets, respectively. They have developed a web server to predict PVPs and non-PVPs for any user. Charoenkwan et al. also introduced a scoring card method (SCM) for the identification of PVPs [12]. They have attained an accuracy of 92.52% and 77.66% on training and independent datasets, respectively. They attained a very high accuracy on the training dataset, but the accuracy significantly drops on the independent dataset. Performances of all these methods on the independent dataset drop significantly. This underscores the necessity to develop a computational method that will achieve nearly equal performances on both the training and independent datasets when predicting PVPs and non-PVPs.

In this study, we have introduced a machine learning-based prediction method to predict PVPs and non-PVPs using protein sequence composition features. We have employed well-known basic and ensemble machine learning methods to achieve nearly equal performance on both training and independent datasets and also developed a web server to predict PVPs and non-PVPs for any user. The web server is freely available at http://www.nicedbacteriophagelab.com/ (accessed on 2 January 2023).

## 2. Results

### 2.1. Selection of Features

To achieve nearly equal performances on both training and independent datasets, we tested different combinations of important protein sequence features such as AAC, DPC, PAAC, and CTD. As shown in Table 1 and the full Table in Appendix A, the prediction of most of the features and combinations of features were highly accurate on the training dataset, but accuracy significantly dropped on the independent dataset. Among the sequence features, AAC (20 vector length) and PAAC (50 vector length) features attained nearly equal performance on both the training and independent datasets. Among the combination of sequence features, AAC and DPC (420 vector length) achieved nearly equal performance on both the training and independent datasets. As shown in Table 1 and the full Table in Appendix A, AAC features attained an accuracy of 77% and 83% on training and independent datasets, respectively. PAAC features achieved an accuracy of 83% and 81% on training and independent datasets, respectively. The combination of features AAC and DPC attained an accuracy of 80% and 83% on training and independent datasets, respectively. The above results suggested that performance balance on training and independent datasets is slightly better in a combination of AAC and DPC features. Therefore, we have selected a combination of AAC and DPC features to develop the prediction model.

### 2.2. Selection of Methods

We have employed several basic and ensemble machine learning methods to attain nearly equal performances on both the training and independent datasets. We have reported only the best results for each method using the best parameter set (Appendix A). As shown in Table 1 and Appendix A, all methods attained relatively high performance on the training dataset, but accuracy considerably plummeted on the independent dataset. Most of the basic machine learning methods attained high sensitivity rather than specificity, whereas ensemble learning methods attained a similar type of sensitivity and specificity. Among the basic machine learning methods, RF with AAC features attained nearly equal performances on both the training and independent datasets. RF with AAC features achieved an accuracy of 77%, a precision of 90%, and an f1-score of 84.1% on the training dataset, and an accuracy of 83%, a precision of 97%, and an f1-score of 88.9% on the independent dataset. Among the ensemble learning methods, GBC with AAC and DPC features achieved an accuracy of 80%, a precision of 95%, an f1-score of 86.9% on the training dataset, and an accuracy of 83%, a precision of 97% and an f1-score of 88.9% on the independent dataset. As suggested by Table 2 and Figure 1 and Figure 2, the performance balance between both the training and independent datasets is slightly better with the GBC method. Therefore, we have selected the ensemble machine learning method GBC with AAC and DPC features for developing the final prediction model.

### 2.3. Comparison with Other Methods

The existing methods, such as Ding et al., achieved an accuracy of 85.02% on the training dataset but an accuracy of only 71.3% on the independent dataset [13]. Similarly, Manavalan et al., Tan et al., and Charoenkwan et al. achieved accuracies of 87%, 87.95%, and 92.52% on the training dataset, respectively (shown in Table 3). These above methods achieved accuracies of 79.8%, 75.53%, and 77.66%, respectively, on the independent dataset [12,15,16]. The above results suggested that these methods could achieve high accuracies on the training dataset but are unable to attain similar accuracies on the independent dataset. The proposed method achieved an accuracy of 80% on the training dataset and an accuracy of 83% on the independent dataset. Therefore, in a real-life scenario, such as the independent dataset, our proposed method performed better than other existing methods (shown in Table 3). In addition to this, the performance measures balance between training and independent datasets of the proposed method is better than other existing methods.

### 2.4. Web Server

We have developed a web server to predict PVPs from any phage protein sequence. Therefore, any user can submit and upload their phage protein sequences in FASTA format and obtain results for their submitted protein sequences. We have developed this web server using Django and python and deployed the best prediction model on the web server to predict phage protein sequences such as PVPs or non-PVPs. The submission tab, or page, is the main functional page on this web server. On the submission page, a user can submit a phage protein sequence by either typing or pasting the FASTA protein sequences in the input text box section or by selecting the phage protein sequences (FASTA files) from a computer, mobile, tablet, etc., from the “Input FASTA File” section. For demonstration purposes, we have also provided the “Example” button in the submission tab or page, which will automatically generate a sample of phage protein sequences in the input text box section. We encourage all readers to submit proper FASTA-formatted phage protein sequences. Using the “Submit” button, the user will be redirected to the results page, which will show the submitted phage protein sequence header name, prediction score, and prediction class. The average waiting time is 15–30 s. The submission and result page of this web server is shown in Figure 3 and Figure 4. The step-by-step user guide can also be found in the “Help” tab on the web server.

## 3. Discussion

The identification of PVPs is critical in understanding the relationship between phages and their host bacteria and might play a vital role in the development of novel antibacterial drugs. Phage-driven antibacterial drugs are considered more specific to their bacterial targets, whereas antibiotics target a broad spectrum of both pathogenic and nonpathogenic microorganisms. So far, no side effects have been observed in any of the phage-driven antibacterial drugs [18,19]. Therefore, researchers are more focused on phage-driven antibacterial drugs as an alternative to antibiotics, especially where AMR is a problem [7,20,21]. Phage-driven proteins such as holins, endolysins, and exopolysaccharides are found to have broad prospects in the development of antibacterial drugs [8,22]. Similarly, phage virion proteins (PVPs) might also play a key role in the development of novel antibacterial drugs.

There are several experimental techniques for the identification of PVPs, which are very expensive and strenuous, as well as time consuming. However, several low-cost, less-time-consuming, and less-effort computational approaches have been proposed for the identification of PVPs. In this study, we have offered a machine learning-based prediction method to predict PVPs and non-PVPs. To compare the prediction performance of our proposed method with other existing methods, we have used the same benchmark datasets that have been used previously by most of the existing methods. One of the prime objectives of the computational prediction method is to achieve good and nearly equal performances on both the training and independent datasets. However, the majority of the existing computational methods used to predict PVPs performed very well on the training dataset but were unable to attain similar types of performance on the independent dataset (shown in Table 3). Our proposed best prediction model achieved nearly equal and good performances on both the training and independent datasets. Additionally, we have also executed our method on the Charoenkwan et al. version 2 dataset, which consists of 250 PVPs and 250 non-PVPs in the training dataset, and 63 PVPs and 63 non-PVPs in the independent dataset [17]. We have observed nearly similar types of performance for both the benchmark dataset and the Charoenkwan et al. version 2 dataset (shown in Appendix A). As shown in Table 1, the primary protein sequence features, such as AAC and DPC, are strong enough to differentiate PVPs and non-PVPs. Among the basic machine learning methods, RF performed better than other methods. Likewise, GBC performed better than other ensemble learning methods (shown in Table 1). As shown in Table 1 and the full Table in Appendix A, the performance of RF on AAC and the performance of GBC on ACC and DPC are similar. However, the performance balance (nearly equal sensitivity and specificity) in terms of sensitivity and specificity of the GBC method is better than RF (shown in Appendix A). Therefore, we selected GBC for the final prediction model. Among the existing methods, only a few methods are available publically to predict PVPs for any user. We have developed a user-friendly web server and made it freely available to all users to predict PVPs and non-PVPs from phage protein sequences. The performance and performance balance of our prediction model on both the training and independent datasets is better than the existing methods.

## 4. Materials and Methods

### 4.1. Dataset

We used the benchmark datasets of 99 PVPs and 208 non-PVPs for the training dataset, and 30 PVPs and 64 non-PVPs for the independent dataset. The majority of the existing computational methods for the prediction of PVPs and non-PVPs have used these datasets [12,13,14,15,16]. Originally, these datasets were prepared from the Universal Protein Knowledgebase (UniProtKB) [23]. The subcellular locations of phage proteins have been used to determine PVP and non-PVP. Phage proteins are considered PVP and non-PVP when the subcellular location is virion or not, respectively. Homology threshold values of less than 40% were considered for removing the homologous PVPs and non-PVPs.

### 4.2. Features

Protein primary sequence features, such as amino acid composition (AAC), dipeptide composition (DPC), and sequence-derived features such as pseudo amino acid composition (PAAC) and composition–transition–distribution (CTD) have been found as critical features for the differentiation between diverse groups of proteins [15,24,25,26,27]. We have computed all of these features using the python package “protlearn”. The detailed workflow of this study is shown in Figure 5.

### 4.3. Classification

The differentiation between PVPs and non-PVPs can be viewed as a two-class (binary) classification problem. Therefore, to differentiate between PVPs and non-PVPs, we employed basic machine learning methods such as logistic regression (LR), gaussian Naïve Bayes (GNB), K-nearest neighbor (KNN), decision tree (DT), support vector machine (SVM) and random forest (RF). For the same purpose, we also utilized ensemble learning methods such as the AdaBoost classifier (ABC) and gradient boosting classifier (GBC).

#### 4.3.1. Logistic Regression (LR)

LR is one of the popular machine learning algorithms for binary classification problems. It predicts the outcome variable by analyzing the relationship between one or more independent variables. For hyperparameter tuning, we mainly used the penalty and cost (C) parameters of LR.

#### 4.3.2. K-Nearest Neighbor (KNN)

KNN is one of the simplest nonparametric machine learning algorithms for regression and classification problems. It assumes similarity between the new data point and available data points and assigns the new data point to the class that is most similar to the available classes. We used the n_neighbors, power (p), and weights parameters of KNN for hyperparameter tuning.

#### 4.3.3. Decision Tree (DT)

DT is one of the most powerful and popular nonparametric machine learning algorithms for regression and classification problems. It is a tree-structured-based classifier, where branches represent the decision rules, internal nodes represent the features of a dataset, and each leaf node represents the outcome. For hyperparameter tuning, we used max_depth, min_samples_leaf, and min_samples_split parameters of DT.

#### 4.3.4. Support Vector Machine (SVM)

SVM is one of the well-known and extensively used machine learning algorithms for regression and classification. The SVM classifier explicitly maps the data over a vector space to find a decision surface that maximizes the margin between data points of two or more classes. We utilized the kernel, cost (C), and gamma parameters of SVM for hyperparameter tuning.

#### 4.3.5. Random Forest (RF)

RF is also one of the well-known and extensively used machine learning algorithms for regression and classification. It “grows” several decision trees (DTs) simultaneously, where each node uses a random subset of the features. Each tree in a random forest classifies the new object and “votes” for that class. The forest elects the classification based on a majority vote (over all the trees in the forest). For hyperparameter tuning, we utilized the n_estimators and random_state parameters of RF.

#### 4.3.6. AdaBoost Classifier (ABC)

AdaBoost or adaptive boosting is one of the popular ensemble machine learning algorithms for classification. It builds a strong classifier by combining multiple poorly performing classifiers to obtain a highly accurate and strong classifier. We utilized the n_estimators and random_state parameters of ABC for hyperparameter tuning.

#### 4.3.7. Gradient Boosting Classifier (GBC)

GBC is also one of the well-known ensemble machine learning algorithms for classification. It combines many weak learning models to create a strong predictive model. In GBC, each predictor tries to improve on its predecessor by reducing the losses. GBC models are becoming popular because of their effectiveness at classifying complex datasets. For hyperparameter tuning, we utilized the n_estimators and random_state parameters of GBC.

We utilized the scikit-learn python package for all of the abovementioned classifiers [28]. We used a cutoff of 0.5 and the best parameter set of each machine learning method to identify PVPs (Appendix A).

### 4.4. 10-Fold Cross-Validation

We used 10-fold cross-validation techniques to avoid performance bias in the prediction methods. In 10-fold cross-validation, the dataset is split into 10 equally (or nearly equally) sized segments or folds. Consequently, 10 times of training and testing were conducted; each time a distinct fold of the data is held-out for testing, the remaining 9 folds are used for training. The overall performance of a method was measured using the average performance over 10 folds [29].

### 4.5. Performance Measures

All performance measures of the binary classification problem such as sensitivity, specificity, accuracy, positive predictive value (PPV or precision), Mathew’s correlation coefficient (MCC), and F1 score were calculated using the following equations:(1)$$Sensitivity=\frac{TP}{TP+FN}\times 100\%\;$$
(2)$$Specificity=\frac{TN}{TN+FP}\times 100\%$$
(3)$$Accuracy=\frac{TP+TN}{TP+FP+TN+FN}\times 100\%$$
(4)$$PPV=\frac{TP}{TP+FP}\times 100\%$$
(5)$$MCC=\frac{TP\times TN-FP\times FN}{\surd \left(\;\left(TP+FP\right)\times \left(TP+FN\right)\times \left(TN+FP\right)\times \left(TN+FN\right)\right)}\;$$
(6)$$F1=2\times \frac{Sensitivity\times PPV}{Sensitivity+PPV}\times 100\%$$
where **True Positive (TP)** are PVPs that are correctly identified as PVPs, **False Positive (FP)** are non-PVPs incorrectly identified as PVPs, **True Negative (TN)** are non-PVPs correctly identified as non-PVPs, and **False Negative (FN)** are PVPs incorrectly identified as non-PVPs.

The area under the receiver operating characteristic curve (AUC) was also computed for all cases. 

## 5. Conclusions

In this study, we developed a machine learning-based prediction method to predict PVPs and non-PVPs using phage protein sequences. The primary protein sequence features such as AAC and DPC as well as derived sequence features such as PAAC and CTD were utilized for this study. We employed well-known basic and ensemble machine learning methods to achieve nearly equal and good performances on both the training and independent datasets. The best prediction model was deployed to our web server to predict PVPs from phage protein sequences for any users. The web server might be useful for the large-scale screening of PVPs from uncharacterized protein sequences of bacteriophages. Due to the availability of the small number of experimentally verified PVPs and non-PVPs in the public domain, the performance of prediction methods of PVPs and non-PVPs is slightly inferior compared to the prediction methods developed in the other fields of biomedical research. The availability of large numbers of experimentally verified PVPs and non-PVPs will help us to develop a better prediction method in the future.

## Figures and Tables

**Figure 1 molecules-28-02238-f001:**
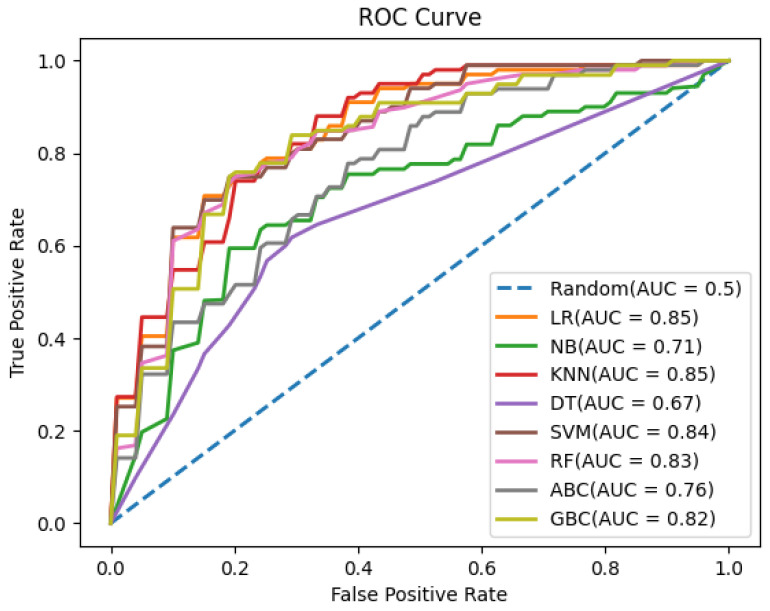
AUC for different machine learning methods using the training dataset and the AAC and DPC features.

**Figure 2 molecules-28-02238-f002:**
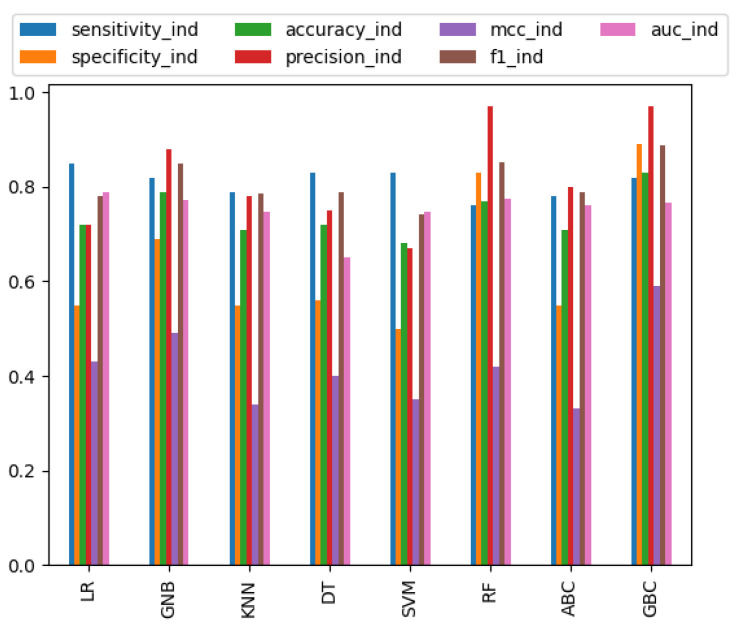
Performance measures for different machine learning methods using the independent dataset and the AAC and DPC features.

**Figure 3 molecules-28-02238-f003:**
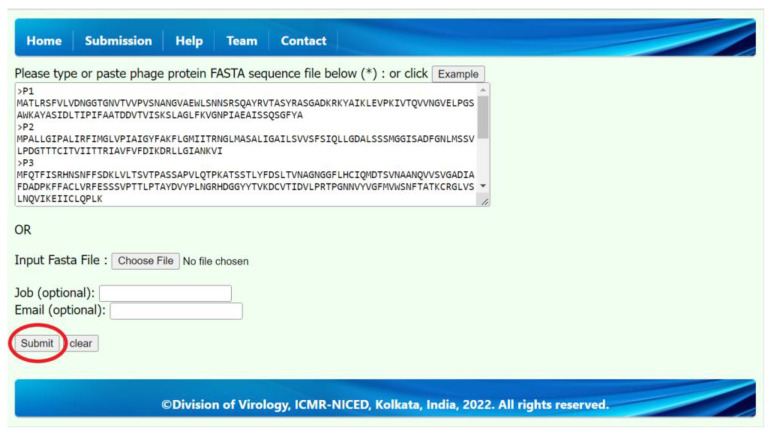
The submission page of the web server. Example phage protein sequences are also shown in the text field.

**Figure 4 molecules-28-02238-f004:**
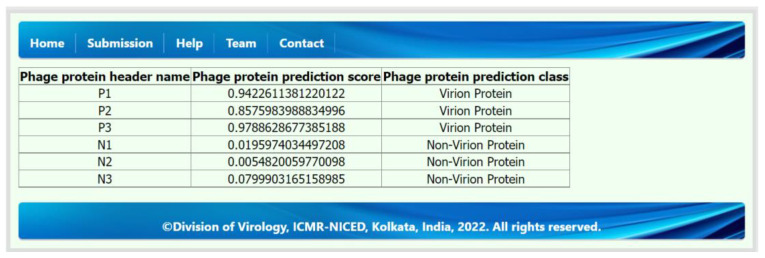
The result page of the web server shows a sample of example phage protein sequence prediction results in table format.

**Figure 5 molecules-28-02238-f005:**
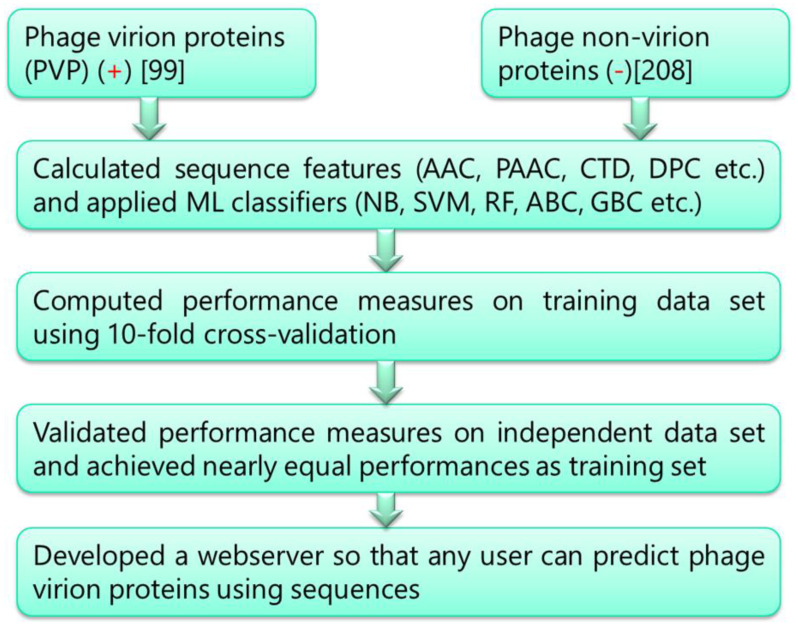
Workflow of the study.

**Table 1 molecules-28-02238-t001:** Features and methods-wise performance measures on the training and the independent datasets using 10-fold cross-validation.

Sequence Features	Features Length	Method	Training Dataset			Independent Dataset		
			accuracy	precision	f1	accuracy_ind	precision_ind	f1_ind
Amino acid composition (AAC)	20	SVM	0.77	0.76	0.82	0.71	0.7	0.768
		RF	0.77	0.9	0.841	0.83	0.97	0.889
		ABC	0.67	0.86	0.784	0.7	0.78	0.78
		GBC	0.8	0.9	0.864	0.74	0.86	0.824
Dipeptide composition (DPC)	400	SVM	0.87	0.86	0.903	0.72	0.77	0.794
		RF	0.67	0.95	0.799	0.73	0.97	0.833
		ABC	0.8	0.86	0.86	0.73	0.78	0.799
		GBC	0.73	0.9	0.824	0.76	0.94	0.84
Pseudo amino acid composition (PAAC)	50	SVM	0.87	0.86	0.903	0.69	0.72	0.762
		RF	0.77	0.9	0.841	0.77	0.97	0.852
		ABC	0.8	0.95	0.869	0.77	0.84	0.83
		GBC	0.83	0.95	0.886	0.81	0.94	0.87
Composition–transition–distribution (CTD)	343	SVM	1	1	1	0.69	0.84	0.787
		RF	0.77	1	0.857	0.71	0.95	0.819
		ABC	0.87	0.95	0.908	0.63	0.78	0.738
		GBC	0.87	0.95	0.908	0.66	0.78	0.759
AAC and DPC	420	SVM	0.83	0.81	0.87	0.68	0.67	0.741
		RF	0.67	0.95	0.799	0.77	0.97	0.852
		ABC	0.7	0.81	0.789	0.71	0.8	0.79
		GBC	0.8	0.95	0.869	0.83	0.97	0.889
AAC, DPC, and CTD	763	SVM	1	1	1	0.69	0.84	0.787
		RF	0.77	0.95	0.851	0.78	1	0.857
		ABC	0.87	0.95	0.908	0.72	0.81	0.8
		GBC	0.8	0.95	0.869	0.73	0.92	0.826
AAC, DPC, PAAC, and CTD	813	SVM	0.97	0.95	0.974	0.76	0.83	0.825
		RF	0.73	0.95	0.832	0.74	0.97	0.84
		ABC	0.87	0.95	0.908	0.72	0.81	0.8
		GBC	0.8	0.95	0.869	0.79	0.91	0.857

**Table 2 molecules-28-02238-t002:** Performance measures of the AAC and DPC features using the training and the independent datasets.

Method	Training Dataset					Independent Dataset				
	accuracy	precision	mcc	f1	auc	accuracy_ind	precision_ind	mcc_ind	f1_ind	auc_ind
LR	0.83	0.86	0.62	0.88	0.847	0.72	0.72	0.43	0.78	0.789
GNB	0.73	0.9	0.29	0.824	0.714	0.79	0.88	0.49	0.849	0.772
KNN	0.87	0.9	0.68	0.9	0.848	0.71	0.78	0.34	0.785	0.746
DT	0.77	0.86	0.43	0.84	0.65	0.72	0.77	0.39	0.794	0.654
SVM	0.83	0.81	0.65	0.87	0.842	0.68	0.67	0.35	0.741	0.746
RF	0.67	0.95	−0.1	0.799	0.828	0.77	0.97	0.42	0.852	0.774
ABC	0.7	0.81	0.26	0.789	0.759	0.71	0.8	0.33	0.79	0.762
**GBC**	**0.8**	**0.95**	**0.49**	**0.869**	**0.822**	**0.83**	**0.97**	**0.59**	**0.889**	**0.768**

**Table 3 molecules-28-02238-t003:** Performance comparisons between the proposed method and other existing methods.

Method	Training Dataset				Independent Dataset			
	Accuracy (%)	Sensitivity (%)	Specificity (%)	MCC	Accuracy_ ind (%)	Sensitivity_ ind (%)	Specificity_ ind (%)	MCC_ind (%)
Feng et al., 2013 [14]	79.15	75.76	80.77	-	-	-	-	-
Ding et al., 2014 [13]	85.02	75.76	89.42	-	71.30	60.00	76.50	0.357
Manavalan et al., 2018 [15]	87.00	73.70	93.30	0.695	79.80	66.70	85.90	0.531
Tan et al., 2018 [16]	87.95	83.83	89.90	0.761	75.53	70.00	78.13	0.464
Charoenkwan et al., 2020 [17]	92.52	95.89	90.86	0.849	77.66	76.67	78.13	0.523
**Proposed** **Method**	**80.00**	**80.00**	**80.00**	**0.490**	**83.00**	**82.00**	**89.00**	**0.590**

## Data Availability

The web server is freely available at http://www.nicedbacteriophagelab.com/ accessed date on 2 January 2023. Source codes are available at https://github.com/ranjan1010/PhageVirionProteinPrediction.git (accessed date on 2 January 2023).

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
