# Peer review of "Prediction of Phage Virion Proteins Using Machine Learning Methods"

_molecules, 2023, doi:10.3390/molecules28052238_

Round 1

Reviewer 1 Report (Previous Reviewer 2)

The authors replied "As suggested by the reviewer we will incorporate these tables in revised manuscript", but I cannot find them in either the main text or supplementary data. In the data set section, the description of the new data set (newer and larger one) is not included, either. 

There are still several grammatical errors throughout the manuscript. For example, in line 178, "is achieved" should be "achieved"; in line 68, "is significantly drops" should be "significantly drops". Such grammatical errors should be fixed completely before the manuscript is accepted. 

Author Response

Reviewer 2 Report (New Reviewer)

In this paper, the author proposed a machine-learning method to predict the phage virion proteins using phage protein sequences. They significantly improved the accuracy of independent datasets and narrowed the gap in accuracy between the training dataset and independent datasets by selecting a more suitable sequence feature and prediction model. Moreover, a user-friendly web server designed by the author may develop a better prediction on large numbers of PVPs.

There are some problems, which must be solved before it is considered for publication. If the following problems are well-addressed, this reviewer believes that the essential contribution of this paper is important for the prediction of phage virion proteins.

  1. As can be seen in Fig1, the RF method on AAC has the same accuracy on the independent dataset with GBC methods on AAC and DPC, why the author choose the latter to develop the final prediction model.
  2. The accuracy of the training dataset is still much lower than other groups, like Chroenkwan, do the author think they can improve the accuracy of training data? If that happens, will the accuracy of the independent dataset also have equal performance?
  3. The formula between line 228 and 233 are garbled, the author should correct it.
  4. Authors need to unify the format of all references, such as whether to use both the starting and ending page numbers, and whether to abbreviate the journal name.

Author Response

This manuscript is a resubmission of an earlier submission. The following is a list of the peer review reports and author responses from that submission.

Round 1

Reviewer 1 Report

This paper tried to predict PVPs and non-PVPs by protein sequence composition features. To achieving comparable performance in the independent sets as in training data, the authors tested basic and ensembled machine learning methods.

1. The authors selected 99 PVPs and 208 non-PVPs as training data, 30 PVPs and 64 non-PVPs for independent data, respectively. Are these data accepted benchmark data? Are there any other gold standard data?

2. Were the features used in this paper different from previous literatures? The authors should discuss the merits and biological meanings of these features.

3. The detailed information of each machine learning methods such as the parameters or cutoffs should be listed.

4. The authors just compared their best results with previous literatures. However, the authors should make it more clearly whether the comparison based on the same data.

5. This paper simply listed the results of running the classical machine learning methods. The authors should highlights their contributions on algorithms, feature selections and so on.  

Author Response

Response to Reviewer 1 Comments

This paper tried to predict PVPs and non-PVPs by protein sequence composition features. To achieving comparable performance in the independent sets as in training data, the authors tested basic and ensembled machine learning methods.

Point 1: The authors selected 99 PVPs and 208 non-PVPs as training data, 30 PVPs and 64 non-PVPs for independent data, respectively. Are these data accepted benchmark data? Are there any other gold standard data?

Response 1: Yes, as mentioned in the Dataset section in the revised manuscript, this is the benchmark dataset for the prediction of phage virion proteins (PVPs) and non-PVPs. The majority of existing computational methods have been used in this dataset. Therefore, we have utilized this dataset for developing our prediction model as well as performance comparison with the other prediction methods. (Page no. 12 and line no. 202-205 in the revised manusript)

Point 2: Were the features used in this paper different from previous literatures? The authors should discuss the merits and biological meanings of these features.

Response 2: Please note that the primary goal of the study is to predict PVPs from protein sequences and build a webserver so that any user can identify PVPs and non-PVPs for their phage proteins. Hence, we searched for simple sequence features, which could efficiently identify PVPs. The extensive literature survey suggests that protein primary sequence features such as amino acid composition (AAC), di-peptide composition (DPC), and sequence-derived features such as pseudo amino acid composition (PAAC), composition-transition-distribution (CTD) have been found as important features for the differentiation among the various groups of proteins. We found that the simple sequence features such as AAC and DPC can efficiently differentitate between PVP and non-PVPs. (Page no. 12-13 and line no. 211-214 in the revised manusript)   

Point 3: The detailed information of each machine learning methods such as the parameters or cutoffs should be listed.

Response 3: Kindly note that we have used a cutoff of 0.5 and the best parameter set of each machine learning method. We have used the grid search method in order to find the best parameter set for each machine-learning method. We have found that the best parameters (penalty = l2 and cost (C) = 100) for Logistic regression (LR), (n_neighbors = 20, p = 1, weights = distance) for K-Nearest Neighbor (KNN), (max_depth = 10, min_samples_leaf = 2, min_samples_split = 4) for Decision Tree (DT), (C = 10, gamma = 0.1, kernel = rbf) for Support Vector Machine (SVM), (n_estimators = 100, random_state = 42) for Random Forest (RF), AdaBoost classifier (ABC) and Gradient Boosting classifier (GBC). As suggested by the reviewer, we have included the best parameters of each method in the supplementary Table S2. (Page no. 14 and line no. 226-228 in the revised manusript)

Point 4: The authors just compared their best results with previous literatures. However, the authors should make it more clearly whether the comparison based on the same data.

Response 4: Yes, the performance comparison of our method with previous literature is based on the same data, which is widely used for the prediction of PVPs and non-PVPs. We have compared the best result of our method with the best results of previous literature. (Page no. 12 and line no. 202-205 in the revised manusript)

Point 5: This paper simply listed the results of running the classical machine learning methods. The authors should highlights their contributions on algorithms, feature selections and so on.  

Response 5: Kindly note that as mentioned in the revised manuscript, we have used classical and ensemble machine learning methods to solve the binary classification problem. The primary goal for applying these machine learning methods is to achieve nearly equal performance for identifying PVPs and non-PVPs on both the training and independent datasets.  For this purpose, we have also used greedy search feature selection methods. We found that most of the features and combinations of features attained high accuracy on the training dataset but accuracy are significantly dropped on the independent dataset. Among the sequence features AAC (20 vector length) and PAAC (50 vector length) features attained nearly equal performance on both the training and independent datasets. Among the combination of sequence features, AAC and DPC (420 vector length) achieved nearly equal performance on both the training and independent datasets. As shown in Table 1 and the full Table in Supplementary Table S1, AAC features attained an accuracy of 77% and 83% on training and independent datasets, respectively. PAAC features achieved an accuracy of 83% and 81% on training and independent datasets, respectively. The combination of features AAC and DPC attained an accuracy of 80% and 83% on training and independent datasets, respectively. The above result suggested that performance balance on training and independent datasets is slightly better in a combination of AAC and DPC features. Therefore, we have selected a combination of AAC and DPC features to develop the prediction model. (Page no. 5 and line no. 93-109 and Page no. 13 and line no. 219-225 in the revised manusript)

Reviewer 2 Report

The authors proposed a machine learning predictor to identify phage virion proteins which yields better predictive performance compared to existing methods. The proposed method is quite simple and straightforward with little novelty. Common features such as AAC, DPC, PAAC, CTD, and the combinations of them are used; common predictors such as SVM, RF, and several boosting models have been tested on different combinations of feature sets. 

The data set used in this study was published years ago, consisting of 99 PVPs and 208 non-PVPs for training, and 30 PVPs and 64 non-PVPs for testing. However, a new data set published in 2020 (by Charoenkwan) consists of 250 PVPs and 250 non-PVPs for training, and 63 PVPs and 63 non-PVPs for testing. The new data set has much more PVPs in the training set, which can benefit the training of machine learning models, leading to improved predictive capability. I wonder why the authors use an outdated data set. It is strongly suggested the comparison is made on the new test data set.

It is also suggested the authors compare their model to SCORPION, a method published earlier this year on Scientific Reports.  

The proposed method has little novelty in terms of features and machine learning models and yet it achieves better performance than the compared methods. For example, Manavalan et al. use a more sophisticated feature selection algorithm based on feature importance score and achieve an MCC of 0.531. The proposed method uses a relatively simple feature engineering approach (different combinations of feature sets) and achieve a much higher MCC of 0.59. Can the authors explain the situation? 

Minor:

The manuscript has a number of grammatical errors and requires extensive English editing.

Round 2

Reviewer 2 Report

The authors have adequately addressed the review comments. Experiments have been carried out on the larger and newer data sets provided by Charoenkwan et al.

The authors point out that the evaluation scores shown in the SCORPION paper may have some erroneous numbers, but, in general, the benchmark results shown in Table 3 and 4 seem to be pretty much correct. I thus encourage the authors include Table 1S and 2S (from the response) in the paper or in the supplementary table. I believe it is important information to the readers how the proposed method perform on the newer and larger data set.